# Exercise Ameliorates Diabetic Kidney Disease in Type 2 Diabetic Fatty Rats

**DOI:** 10.3390/antiox10111754

**Published:** 2021-11-03

**Authors:** Itaru Monno, Yoshio Ogura, Jing Xu, Daisuke Koya, Munehiro Kitada

**Affiliations:** 1Department of Diabetology and Endocrinology, Kanazawa Medical University, Uchinada 920-0293, Ishikawa, Japan; imonno@kanazawa-med.ac.jp (I.M.); namu1192@kanazawa-med.ac.jp (Y.O.); xujing@kanazawa-med.ac.jp (J.X.); koya0516@kanazawa-med.ac.jp (D.K.); 2Division of Anticipatory Molecular Food Science and Technology, Medical Research Institute, Kanazawa Medical University, Uchinada 920-0293, Ishikawa, Japan; 3Omi Medical Center, Kusatsu 525-8585, Shiga, Japan

**Keywords:** diabetic kidney disease, exercise, inflammation, oxidative stress, autophagy, AMP-activated kinase, mechanistic target of rapamycin complex 1

## Abstract

Lifestyle improvement, including through exercise, has been recognized as an important mode of therapy for the suppression of diabetic kidney disease (DKD). However, the detailed molecular mechanisms by which exercise exerts beneficial effects in the suppression of DKD have not yet been fully elucidated. In this study, we investigate the effects of treadmill exercise training (TET) for 8 weeks (13 m/min, 30 min/day, 5 days/week) on kidney injuries of type 2 diabetic male rats with obesity (Wistar fatty (fa/fa) rats: WFRs) at 36 weeks of age. TET significantly suppressed the levels of albuminuria and urinary liver-type fatty-acid-binding protein (L-FABP), tubulointerstitial fibrosis, inflammation, and oxidative stress in the kidneys of WFRs. In addition, TET mitigated excessive apoptosis and restored autophagy in the renal cortex, as well as suppressed the development of morphological abnormalities in the mitochondria of proximal tubular cells, which were also accompanied by the restoration of AMP-activated kinase (AMPK) activity and suppression of the mechanistic target of rapamycin complex 1 (mTORC1). In conclusion, TET ameliorates diabetes-induced kidney injury in type 2 diabetic fatty rats.

## 1. Introduction

The number of patients with type 2 diabetes mellitus (T2DM) has been increasing worldwide. Vascular complications due to diabetes include microvascular disorders, including retinopathy, neuropathy, and kidney disease, and macrovascular disorders caused by atherosclerosis [1]. Among them, diabetic kidney disease (DKD) occurs in approximately 30–40% of diabetic patients and is still the leading cause of end-stage kidney disease (ESKD) [2]. Therefore, establishing effective treatments for DKD, in order to preserve kidney function, is necessary. The well-recognized pathogenesis of DKD includes increased hyperglycemia, along with chronic inflammation and oxidative stress [3,4]. In addition, recent accumulating evidence from numerous basic studies has indicated the involvement of novel pathways, including excess apoptosis and impaired adequate autophagy [5]. Previously, we have reported that mechanistic target of rapamycin complex 1 (mTORC1) activation-related autophagy impairment is involved in the accumulation of abnormal mitochondria and excess apoptosis in the renal proximal tubular cells of Wistar fatty (fa/fa) rats, an animal model for T2DM with obesity [6]. Therefore, improvements in the inflammation, oxidative stress, apoptosis, and autophagy status should be a therapeutic target for suppressing diabetic kidney injury.

Lifestyle interventions serve as important basic treatments for patients with T2DM and DKD. Among the various lifestyle interventions, increasing physical activity or regular exercise is generally a widely accepted method for the improvement of health outcomes, including overall well-being, quality of life, and cardiometabolism [7]. In CKD patients (including those with DKD), several epidemiologic/prospective studies and systematic reviews with limited randomized controlled trials (RCTs) have demonstrated the benefits of increasing physical activity or regular exercise on the reduction of mortality risk, improvement of exercise tolerance metabolic health, or suppression of kidney disease progression [8,9,10,11,12,13]. Therefore, exercise has been recommended as a lifestyle intervention for the suppression of DKD, as per the KDIGO guideline 2020 [14]. One of the mechanisms of exercise is renoprotection: physical exercise is shown to have a protective effect against renal damage in diabetic animals, through anti-inflammatory and anti-oxidative stress effects [15,16,17]. However, what is still unclear is whether physical exercise exhibits a renoprotective effect in advanced DKD, as well as details of the molecular mechanisms.

For this study, we investigated whether regular physical exercise using a treadmill may exert renoprotective effects, focusing on the alteration of anti-inflammatory/oxidative stress, apoptosis, and autophagy in Wistar fatty (fa/fa) rats.

## 2. Materials and Methods

### 2.1. Antibodies

Kidney injury molecule 1 (KIM-1) antibody was purchased from R&D Systems, Inc. (Minneapolis, MN, USA). Cluster of differentiation 68 (CD68) antibody was purchased from Serotec (Oxford, UK). Nitrotyrosine antibody was purchased from Trans Genic Inc. (Hyogo, Japan). Nuclear factor-erythroid 2-related factor 2 (Nrf2) antibody was purchased from Abcam (Cambridge, MA, USA). Heme oxygenase 1 (HO-1), cleaved caspase 3, phospho-S6 ribosomal protein (Ser 235/236) (p-S6RP), S6RP, phospho-AMP-activated protein kinase (Thr172) (p-AMPK), and AMPK antibodies were obtained from Cell Signaling Technology (Beverly, MA, USA). p62/SQSTM1 (p62) was obtained from Medical & Biological Laboratories (Tokyo, Japan). β-actin was purchased from Sigma-Aldrich (Saint Louis, MO, USA).

### 2.2. Experimental Animals

The study was approved by the Research Center for Animal Life Science of Kanazawa Medical University. All experiments were performed in accordance with the principles of laboratory animal care. Male and female Wistar lean (fa/+) rats (WLRs; Takeda Pharmaceutical Company Biological Institute, Osaka, Japan) were maintained in temperature-controlled (23 ± 1 °C) rooms on a 12-h light/dark cycle with free access to water and chow, as previously described [6].

### 2.3. Experimental Protocol

At 34 weeks of age, male non-diabetic WLRs and male diabetic Wistar fatty (fa/fa) rats (WFRs) were randomly divided into three groups: (1) sedentary WLRs (WLRs, *n* = 8), non-diabetic and non-exercise control; (2) sedentary WFRs (WFRs, *n* = 8); and (3) WFRs with treadmill exercise training (TET; WFRs-T, *n* = 7). Rats in the WFRs-T group were allowed to run on a 0% gradient treadmill (KN-73, Natsume Seisakusho, Tokyo, Japan) [18]. The treadmill speed was gradually increased, over 2 weeks, to 13 m/min (10 min/day, 3 days/week) for the acclimatization to exercise. At 36 weeks of age, TET (13 m/min, 30 min/day, 5 days/week) without exhaustion was performed for 8 weeks. The intensity of the TET in this study is equivalent to approximately 50–60% of the maximal oxygen consumption (VO_2max_), as previously reported [19,20]. Body weight, food consumption, and blood glucose were measured every four weeks. Sedentary WLRs and WFRs were placed on the nonmoving treadmill for 30 min a day, 5 days a week, as a non-exercise control. At 44 weeks of age, urine samples of individual rats were collected using metabolic cages [6]. Rats were anesthetized with isofluran, and the kidneys were subsequently removed, as previously reported [6]. At the same time, the perirenal, retroperitoneal, and epididymal fat was dissected respectively, and fat weight was defined as the sum of their fat pads, in this study. Gastrocnemius and soleus muscles were also removed. After 12 h following the end of the exercise session, we collected tissue samples under the not-fasting state.

### 2.4. Biochemical Measurements

HbA1c levels were measured by a DCA 2000 Analyzer (Siemens Medical Solutions Diagnostics, Tokyo, Japan) at the end of the experiment [6,21]. Urinary albumin (NEPHRAT II, Exocell Inc., Philadelphia, PA, USA), liver-type fatty-acid-binding protein (L-FABP; mouse/rat FABP1/L-FABP, R&D Systems, Inc., Minneapolis, MN, USA), which is the one of the biomarkers for renal tubular cell damage and oxidative stress, and urinary creatinine (Creatinine Colorimetric Assay Kit, Cayman Chemical Inc., Ann Arbor, MI, USA) were measured using enzyme-linked immunosorbent assay (ELISA) kits [21].

### 2.5. Morphological Analysis and Transmission Electron Microscopy

Paraffin sections of the kidney were stained with Masson’s trichrome (MT) reagent, and immunohistochemical staining was performed by using KIM-1 antibodies (1:100), CD68 antibodies (1:50), and nitrotyrosine antibodies (1:100) [6,22]. For the quantification of the MT-stained fibrosis area and the KIM-1- or CD68-positive areas in immunohistochemical staining, 10 randomly selected tubulointerstitial areas of the renal cortex per rat were measured using the ImageJ software, as previously reported [6]. The morphology of mitochondria in proximal tubular cells was observed by using transmission electron microscopy [6].

### 2.6. Immunoblot Analysis and Real-Time Polymerase Chain Reaction (PCR)

Immunoblotting was performed by using nitrotyrosine antibodies (1:1000), Nrf2 antibodies (1:1000), HO-1 antibodies (1:1000), cleaved caspase 3 antibodies (1:1000), p62/SQSTM1 antibodies (1:1000), p-S6RP antibodies (1:2000), S6RP antibodies (1:1000), p-AMPK antibodies (1:1000), and AMPK antibodies (1:1000). Isolation of total RNA from the renal cortex, cDNA synthesis, and quantitative real-time PCR were performed [6]. TaqMan probes for type III collagen, Kim-1, Cd68, toll-like receptor 2 (Tlr2), toll-like receptor 4 (Tlr4), tumor necrosis factor-α (Tnf-α), and interleukin-6 (Il-6) were obtained from Thermo Fisher Scientific (Waltham, MA, USA). The analytical data were adjusted to the levels of 18S expression levels as an internal control.

### 2.7. Statistical Analysis

Data are expressed as means ± standard deviation (SD). ANOVA followed by Tukey’s multiple comparison test was used to determine the significance of pairwise differences among the three groups. *p* < 0.05 was considered to indicate significant differences.

## 3. Results

### 3.1. Characteristics of the Experimental Rats

The characteristics of the considered rats at 44 weeks of age are shown in Figure 1. The whole-body and kidney weights and mean food intake were significantly higher for the WFRs than the WLRs. Compared with the WFRs, the WFRs-T exhibited significant weight loss, with reduced fat weight, while there were no differences in kidney weight and mean food intake (Figure 1A–D). Gastrocnemius and soleus muscle weights were significantly reduced in the WFRs compared with the WLRs. However, TET resulted in a significant increase in the gastrocnemius muscle weights of the WFRs-T compared with the WFRs (Figure 1E,F). The fasting blood glucose levels were not significantly different among all groups (Figure 1G). The blood glucose under ad libitum feeding and HbA1c levels were significantly elevated in the WFRs and WFRs-T compared with the WLRs; however, there were no differences between the WFRs and WFRs-T (Figure 1H,I). Levels of urinary albumin and L-FABP excretion were markedly increased in the WFRs compared with the WLRs; however, these increased levels were significantly decreased with TET, i.e., in the WFRs-TET (Figure 1J,K).

### 3.2. Changes in Renal Fibrosis and Tubular Cell Damage

Renal fibrosis, evaluated by MT staining, and collagen III expression in the renal cortex were significantly increased in the kidneys of the WFRs compared with the WLRs (Figure 2A–C). In addition, the intensity of KIM-1 immunohistochemical staining and Kim-1 mRNA expression in the renal cortex were significantly increased in the kidneys of WFRs compared with the WLRs (Figure 2A,D,E). Importantly, TET clearly ameliorated all these renal injuries observed in the WFRs, as observed in the WFRs-T.

### 3.3. Changes in Inflammation in the Kidney

The intensity of CD68 immunohistochemical staining in tubulointerstitial areas (Figure 3A,B) and mRNA expression of inflammation-related genes, including Cd68, Tlr2, Tlr4, Tnf-α, and Il-6 (Figure 3C–G) were markedly higher in the WFRs group than the WLRs group; however, TET significantly ameliorated the renal inflammatory changes in the WFRs. 

### 3.4. Changes in Oxidative Stress in the Kidney

The intensity of nitrotyrosine immunohistochemical staining in the kidney of the WFRs was higher than that of the WLRs. In addition, the nitrotyrosine, Nrf2, and HO-1 protein levels were significantly increased in the renal cortex of the WFRs compared with the WLRs (Figure 4A–G). TET clearly ameliorated all oxidative stress-related molecules in the WFR kidneys. 

### 3.5. Changes in Mitochondrial Morphology, Apoptosis, and Autophagy in the Kidney

Mitochondrial morphological changes, including fragmentation or swelling, were observed in the proximal tubular cells of WFRs. TET suppressed the development of all these mitochondrial alterations (Figure 5A). The cleaved caspase 3 and p62/SQSTM1 (p62) protein levels were significantly higher in the renal cortex of the WFRs compared with the WLRs (Figure 5B–E). p62 is recognized as one of the markers of autophagy, and increased p62 expression indicates the impairment of autophagy. TET had clear improvements in mitigating excessive apoptosis and restoring impaired autophagy in the WFRs. Additionally, the levels of the downstream molecule of mechanistic target of rapamycin complex 1 (mTORC1), p-S6RP protein, were significantly increased in the renal cortex of the WFRs compared with WLRs (Figure 5F,G); however, TET significantly decreased the p-S6RP protein levels and elevated p-AMPK protein levels in the renal cortex of the WFRs (Figure 5H,I). Therefore, the improvements in the renal cortex due to TET were accompanied by AMPK activation and mTORC1 suppression.

## 4. Discussion

In this study, TET ameliorated urinary albumin and L-FABP excretion as well as kidney injuries, including tubular cell damage and tubulointerstitial fibrosis, inflammation, and oxidative stress, in WFRs. In addition, TET resulted in improvements by mitigating excessive apoptosis and restoring impaired autophagy in the renal cortex as well as mitochondrial morphology with respect to abnormalities in proximal tubular cells, which were also accompanied by AMPK activation and mTORC1 suppression. The beneficial effects on the kidney of TET were exerted independent of glucose levels, because there was no difference in the levels of HbA1c and blood glucose under ad libitum feeding between sedentary WFRs and WFRs with TET.

Chronic inflammation and oxidative stress play crucial roles in DKD pathogenesis and are closely linked to tubular cell damage and fibrosis [3,4,23]. In this study, the expression of inflammation-related genes and macrophage infiltration were significantly increased in the tubulointerstitial area of WFRs. In addition, the intensity of nitrotyrosine staining in particular tubular cells of WFRs was clearly enhanced, indicating an increase in the formation of peroxynitrite, which consists of nitric oxide (NO) and a superoxide anion (O_2_^−^). Moreover, HO-1 is regulated by Nrf2, a key transcription factor in the regulation of cellular redox balance [24] that functions as one of the defensive molecules against oxidative stress in kidney disease [25]. In this study, the expression of Nrf2/HO-1 was significantly increased in the kidneys of WFRs as an adaptive response, indicating the enhancement of oxidative stress. Furthermore, urinary excretion of L-FABP [6,21,26]—a marker of tubular cell oxidative stress—was significantly increased in the WFRs [6]. TET clearly mitigated both inflammation and oxidative stress in WFR kidneys. Previously, Ishikawa et al. also demonstrated that low-intensity exercise (5 m/min for 30 min, 3 days a week) attenuates the progression of early diabetes-induced kidney injury through the mitigation of oxidative stress and inflammation in type 2 diabetic KK-*A ^y^* mice [17].

Accumulating evidence demonstrates that the reduction in AMPK activity and mTORC1 activation in animals and humans, under caloric excess or diabetic conditions, is involved in the impairment of autophagy [27,28,29,30]. Impaired autophagy is related to the intracellular accumulation of abnormal mitochondria which, in turn, is associated with mitochondrial oxidative stress and excess apoptosis [6,30]. Juszczak et al. reported that exercise training leads to AMPK-mediated autophagy improvement in the kidneys of obese mice induced by high-fat diet [31]. Consistent with these previous studies, our results show both the restoration of AMPK activity and reduced mTORC1 activation as well as the mitigation of both autophagy impairment and abnormal mitochondria accumulation in the proximal tubular cells of WFRs. TET, in the WFRs, resulted in AMPK activation and the suppression of mTORC1 activity, possibly consequently leading to the improvement of oxidative stress and apoptosis through the restoration of autophagy in the kidney. However, further studies are necessary to evaluate whether TET exerts the effects of anti-inflammation/oxidative stress and the reduced excess apoptosis through the restoration of autophagy via AMPK activation and mTORC1 suppression.

In our study, TET reduced fat weights and increased gastrocnemius muscle weights, as observed in the WFRs-T, compared with the WFRs. However, HbA1c levels did not significantly differ between the WFRs with TET and sedentary WFRs. Thus, TET might provide a greater renoprotective effect independent of glucose levels. Previously, Gosh et al. also demonstrated that exercise using a motorized exercise wheel system for 1 h every day in db/db mice improved diabetes-induced kidney injuries including renal cell excess apoptosis, without changes in plasma glucose or insulin status [15]. In our study, results from intraperitoneal glucose and insulin tolerance tests showed no difference between WFRs and WFRs with TET (data not shown). However, we did not evaluate insulin sensitivity by a glucose clamp technic, which has been recognized as a gold-standard method for insulin sensitivity. How exercise can suppress kidney injury through interactions with skeletal muscle and the muscle–kidney crosstalk during exercise has not yet been fully investigated. A previous report showed that Irisin, a myokine, ameliorated tubular cell damage and renal fibrosis in several CKD mice models [32]. In addition, AMPK inhibition decreased the effects of Irisin in myocytes and hepatocytes, possibly indicating that Irisin could be involved in AMPK pathway regulation [33]. Furthermore, Irisin ameliorates high-glucose-induced cardiomyocytes or pressure-overload-induced cardiac hypertrophy and fibrosis through the AMPK/mTORC1 signaling pathway [34,35]. However, the effects of Irisin or other factors on AMPK/mTORC1 regulation in the kidney remain unknown, and elucidation will require further research.

This study had several limitations, as follows: We performed this study using a level of exercise intensity that did not result in the exhaustion of the rats (13 m/min, 30 min/day, 5 days/week). The intensity of exercise was considered to be moderate and approximately 50–60% of the maximal oxygen consumption [20]. However, as we did not evaluate the effect on the kidney according to various levels of exercise intensity, it is unclear how different exercise intensities may affect the kidney. Furthermore, although blood pressure levels may affect the changes in renal function and renal blood flow, we did not collect blood pressure data.

## 5. Conclusions

TET ameliorated functional and histological kidney injuries in WFRs, including albuminuria, increased urinary L-FABP excretion, tubular cell damage, and tubulointerstitial fibrosis, inflammation, and oxidative stress. In addition, TET mitigated excessive apoptosis and restored autophagy in the renal cortex as well as mitochondrial morphology in proximal tubular cells, which was accompanied by AMPK activation and mTORC1 suppression. TET might also provide a greater renoprotective effect independent of glucose levels. However, further studies are necessary to clarify a causality among the TET-induced improvement of inflammation/oxidative stress, decrease in the excess of apoptosis, and the restoration of autophagy through AMPK activation and mTORC1 suppression, in the kidney of WFRs.

## Figures and Tables

**Figure 1 antioxidants-10-01754-f001:**
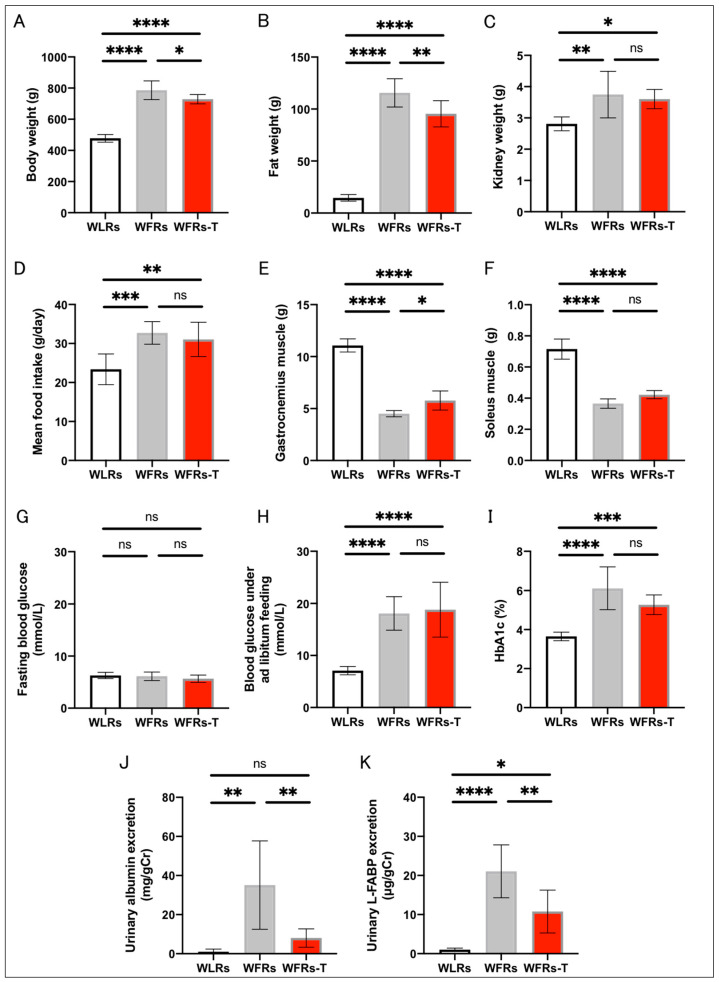
Characteristics of rats at 44 weeks of age: (**A**) whole-body weight; (**B**) fat weight; (**C**) kidney weight; (**D**) mean food intake; (**E**) gastrocnemius muscle weight; (**F**) soleus muscle weight; (**G**) fasting blood glucose; (**H**) blood glucose under ad libitum feeding; (**I**) HbA1c levels (WLRs: *n* = 8, WFRs: *n* = 8, WFRs-T: *n* = 7); (**J**) urinary albumin/creatinine (Cr) ratio; and (**K**) liver-type fatty-acid-binding protein (L-FABP)/Cr, at 44 weeks of age (*n* = 6, respectively). All data are means ± standard deviation (SD). * *p* < 0.05, ** *p* < 0.01, *** *p* < 0.001, **** *p* < 0.0001, ns: not significant. WLRs, sedentary Wistar lean rats; WFRs, sedentary Wistar fatty rats; WFRs-T, Wistar fatty rats with treadmill exercise training.

**Figure 2 antioxidants-10-01754-f002:**
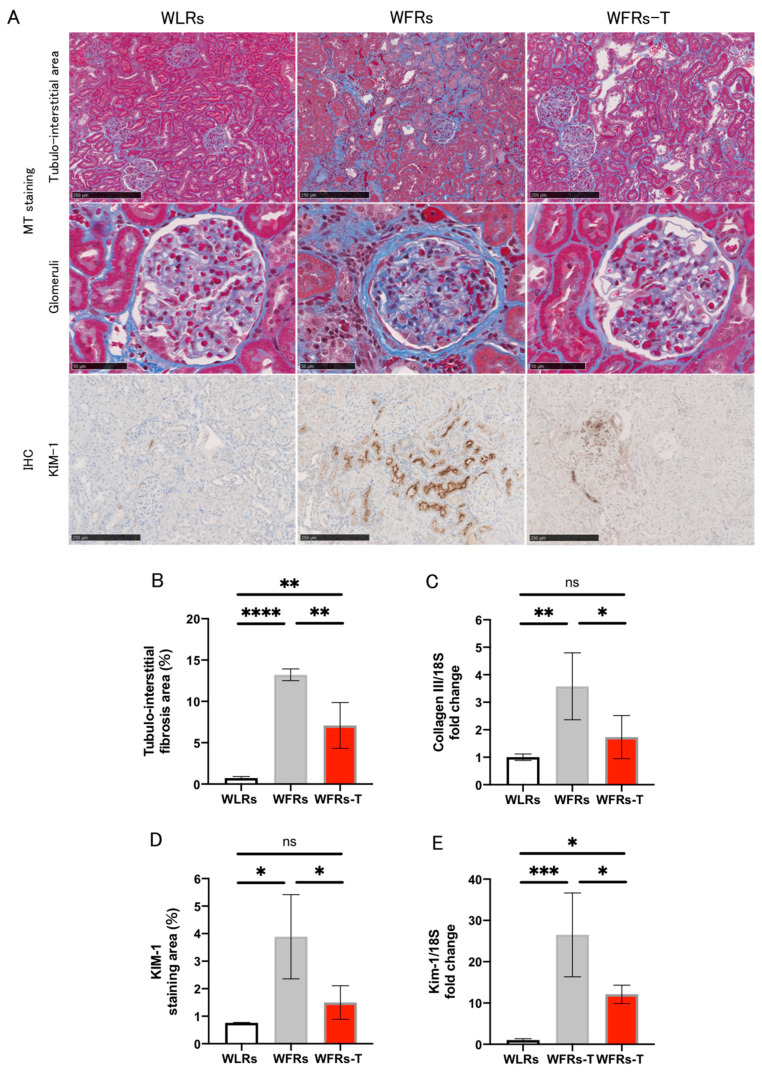
Changes in renal fibrosis and tubular cell damage: (**A**) representative images of MT (Masson’s trichrome) staining in the tubulointerstitial area (scale bar: 250 μm) and glomeruli (scale bar: 50 μm) as well as immunohistochemistry (IHC) of KIM-1 in the tubulointerstitial area (scale bar: 250 μm); (**B**) quantification of tubulointerstitial fibrosis area (*n* = 4); (**C**) mRNA expression of collagen III, normalized according to 18S levels in the renal cortex (*n* = 5); (**D**) quantification of KIM-1 staining area (*n* = 3); and (**E**) mRNA expression of Kim-1, normalized according to 18S levels in the cortex (*n* = 3–4). All data are means ± standard deviation (SD). * *p* < 0.05, ** *p* < 0.01, *** *p* < 0.001, **** *p* < 0.0001, ns: not significant. KIM-1 and Kim-1, kidney injury molecule-1; WLRs, sedentary Wistar lean rats; WFRs, sedentary Wistar fatty rats; WFRs-T: Wistar fatty rats with treadmill exercise training.

**Figure 3 antioxidants-10-01754-f003:**
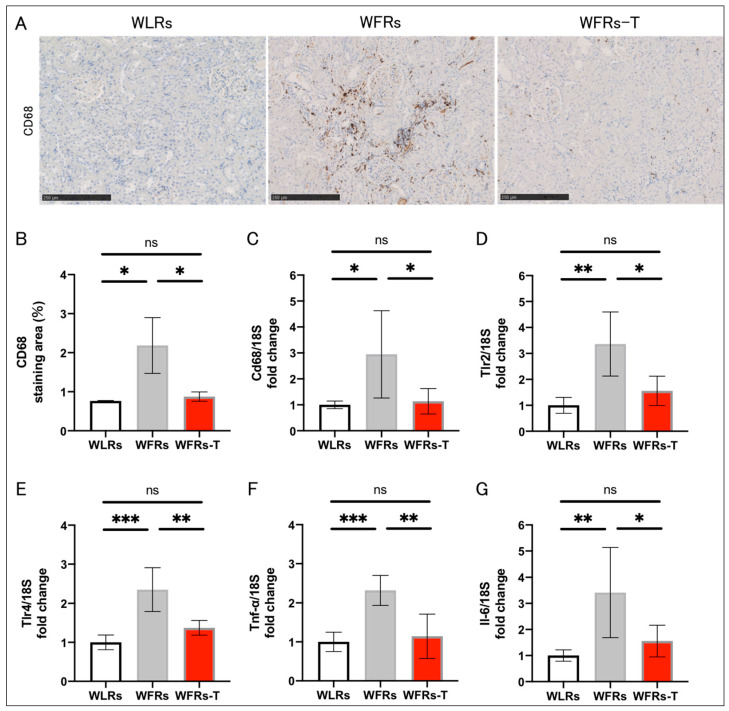
Changes in renal inflammation: (**A**) representative images of CD68 immunohistochemistry (scale bar: 250 μm); (**B**) quantification of CD68 staining area (*n* = 3); and mRNA expression of (**C**) Cd68, (**D**) Tlr2, (**E**) Tlr4, (**F**) Tnf-α, and (**G**) Il-6, adjusted to 18S levels in the cortex (*n* = 5, respectively). All data are means ± standard deviation (SD). * *p* < 0.05, ** *p* < 0.01, *** *p* < 0.001, ns: not significant. Cd68, Cluster of differentiation 68; Tlr2, Toll-like receptor 2; Tlr4, Toll-like receptor 4; Tnf-α, tumor necrosis factor-α; Il-6, interleukin 6; WLRs, sedentary Wistar lean rats; WFRs, sedentary Wistar fatty rats; WFRs-T, Wistar fatty rats with treadmill exercise training.

**Figure 4 antioxidants-10-01754-f004:**
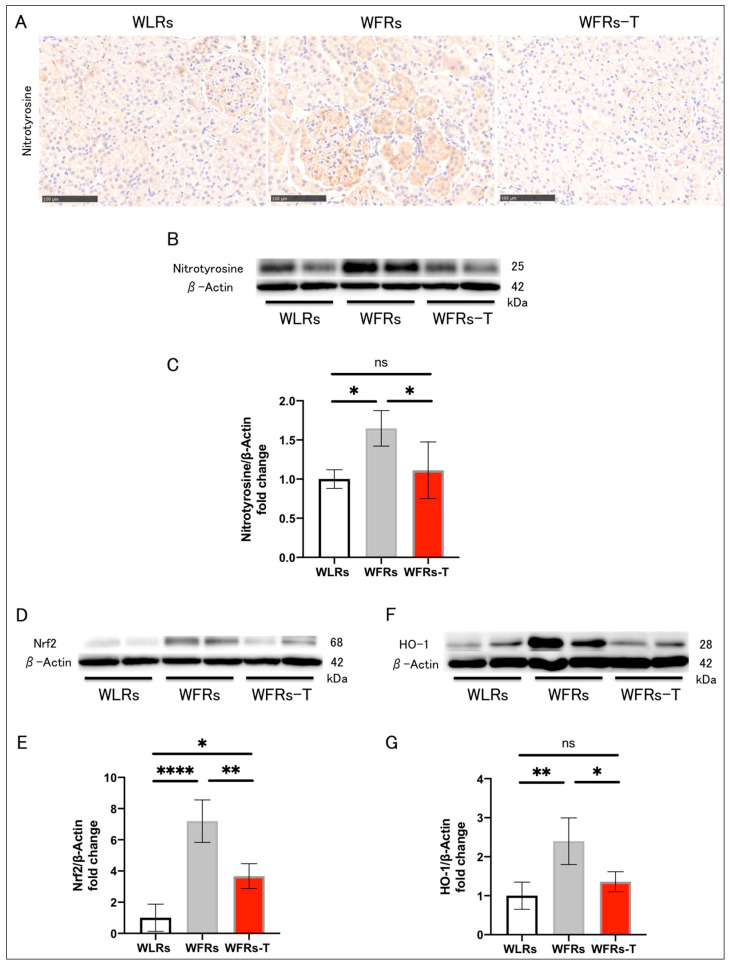
Changes in renal oxidative stress: (**A**) representative images of nitrotyrosine immunohistochemistry (scale bar: 250 μm); (**B**) representative immunoblots of nitrotyrosine and β-actin; (**C**) the quantitative ratios of nitrotyrosine to β-actin in the renal cortex (*n* = 4); (**D**) representative immunoblots of nuclear factor-erythroid 2-related factor 2 (Nrf2) and β-actin; (**E**) the quantitative ratios of Nrf2 to β-actin in the renal cortex (*n* = 4); (**F**) representative immunoblots of heme oxygenase 1 (HO-1) and β-actin; and (**G**) the quantitative ratios of HO-1 to β-actin in the renal cortex (*n* = 4). All data are means ± standard deviation (SD). * *p* < 0.05, ** *p* < 0.01, **** *p* < 0.0001, ns: not significant. WLRs, sedentary Wistar lean rats; WFRs, sedentary Wistar fatty rats; WFRs-T, Wistar fatty rats with treadmill exercise training.

**Figure 5 antioxidants-10-01754-f005:**
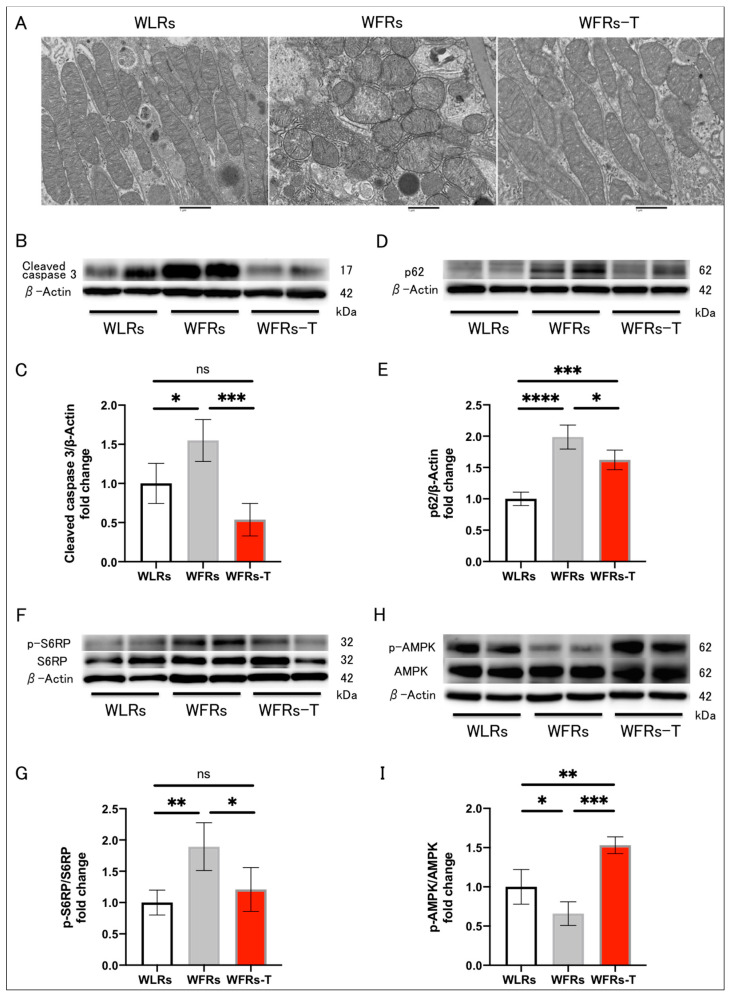
Changes in mitochondrial morphology, apoptosis, and autophagy in the kidney: (**A**) representative transmission electron microscopy images of proximal tubular cells (scale bar: 1 μm); (**B**) representative immunoblots of cleaved caspase 3 and β-actin; (**C**) quantitative ratios of cleaved caspase 3 to β-actin in the renal cortex (*n* = 4); (**D**) representative immunoblots of p62/SQSTM1 (p62) and β-actin; (**E**) quantitative ratios of p62 to β-actin in the renal cortex (*n* = 4); (**F**) representative immunoblots of p-S6RP, S6RP, and β-actin; (**G**) quantitative ratios of p-S6RP to S6RP in the renal cortex (*n* = 4); (**H**) representative immunoblots of p-AMPK, AMPK, and β-actin; and (**I**) quantitative ratios of p-AMPK to AMPK in the renal cortex (*n* = 4). All data are means ± standard deviation (SD). * *p* < 0.05, ** *p* < 0.01, *** *p* < 0.001, **** *p* < 0.0001, ns: not significant. p-S6RP, phispho-S6 ribosomal protein; S6RP, S6 ribosomal protein; p-AMPK, phospho-AMP-activated kinase; AMPK, AMP-activated kinase; WLRs, sedentary Wistar lean rats; WFRs, sedentary Wistar fatty rats; WFRs-T: Wistar fatty rats with treadmill exercise training.

## Data Availability

All the data supporting the findings of this study are included in this article.

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
