# Peer review of "Exercise Ameliorates Diabetic Kidney Disease in Type 2 Diabetic Fatty Rats"

_antioxidants, 2021, doi:10.3390/antiox10111754_

Round 1

Reviewer 1 Report

This is a very interesting and well-designed study. The main limitations are stated in the text by the authors. However, this reviewer missed a more detailed explanation on the absence of differences in the levels of glycated hemoglobin, considering the relevance of this parameter in glucose metabolism 

Author Response

Dear The Editors, Prof. Dr. Javier González-Gallego and Prof. Dr. Jesús R. Huerta

We thank all reviewers and editors for proving us fruitful comments and for the chance to revise our manuscript. We are also thankful to the reviewer’s efforts for spending their valuable time in reading the manuscript. These suggestive comments from reviewers are very convincing. Authors are providing answers to all the comments by point to point raised by reviewers’ comments. In addition, we received English editing by MDPI author services (https://www.mdpi.com/authors/english.).

Reviewer 1

This is a very interesting and well-designed study. The main limitations are stated in the text by the authors. However, this reviewer missed a more detailed explanation on the absence of differences in the levels of glycated hemoglobin, considering the relevance of this parameter in glucose metabolism 

Ans)

We thank reviewer’s evaluation and constructive comments.

The beneficial effects on kidney of TET were exerted in independent glucose levels, because there was no difference on the levels of HbA1c and blood glucose under ad libitum feeding between sedentary WFRs and WFRs with TET. Therefore, we wrote “In our study, TET reduced fat weights and increased gastrocnemius muscle weights, as observed in the WFRs-T, compared with the WFRs. However, HbA1c levels did not significantly differ between the WFRs with TET and sedentary WFRs. Thus, TET might provide a greater renoprotective effect independent of glucose levels.”, in the section of Discussion of first submitted manuscript. In addition, we added the sentences, “Previously, Gosh et al. also demonstrated that exercise using a motorized exercise wheel system for 1 hour every day in db/db mice improved diabetes-induced kidney injuries including renal cell excess apoptosis, without changes in plasma glucose or insulin status. In our study, results from intraperitoneal glucose and insulin tolerance test showed no difference between WFRs and WFRs with TET (data not shown). However, we did not evaluate insulin sensitivity by a glucose clamp technic, which has been recognized as a gold standard method for insulin sensitivity.”, in the section of Discussion. We also showed data regarding IPGTT and IPITT as below (Figure for reviewer). Finally, in the section of Conclusion, we added the sentence, “TET also might provide a greater renoprotective effect independent of glucose levels.”.

Figure for reviewer

Figure for reviewer. IPGTT and IPITT: (a) Data from the intraperitoneal glucose tolerance test (IPGTT) and (c) intraperitoneal insulin tolerance test (IPITT), and (b) AUCs for the IPGTT and (d) IPITT at 44 weeks of age (IPGTT: n=4, IPITT: n=5). All data are means ± standard deviation (SD). ****p < 0.0001, ns: not significant. White circles, WLRs; black circles, WFRs; red triangles, WFRs-T. $$p < 0.01 vs WLRs and WFRs; $$$p < 0.001 vs WLRs and WFRs; \p < 0.05 vs WLRs and WFRs-T; \\p < 0.01 vs WLRs and WFRs-T; \\\p < 0.001 vs WLRs and WFRs-T. WLRs, sedentary Wistar lean rats; WFRs, sedentary Wistar fatty rats; WFRs-T, Wistar fatty rats with treadmill exercise training.

Reviewer 2 Report

In their manuscript by Monno et al. have investigated the ameliorating effect of exercise on diabetes kidney. The in vivo data clearly shows association between exercise, reduced body and fat weight, and renal function. In addition, change in oxidative parameters and glucose sensing proteins (mTOR and AMPK) were observed.

Overall, an interesting study and excellently presented, but some of the conclusions are not properly based on the results.

The authors' main objective was to investigate the molecular mechanism underlying the beneficial effect (abstract) and attributes it to the oxidative environment. However, only association evaluation is presented. An intervention with antioxidants would provide evidence. Here, the author also claims that AMPK activation and mTOR suppression is behind the reduced renal apoptosis (lines 234-240), which is suppressing as mTOR is typically an anti-apoptotic signal (by both main downstream complexes), leading me to a different conclusion -  the alteration in AMPK, mTOR and oxidative markers are the results of the improvement and not mitigate its effect. Perhaps, all observations are due to increased insulin sensitivity?

As DKD is highly complex, and the presented manuscript does advance the knowledge on the subject and the role of oxidative stress, the authors are recommended to revise their manuscript and change all causality to an association (between the reduced oxidative stress, AMPK and mTOR and the improved by TE) in the manuscript (including in the title and abstract). In addition, the discussion should also reflect the need for clarification.

  • Please also add in the abstract and dissection “male rats” or provide evidence that there is no difference in the results due to the gender

Author Response

Dear The Editors, Prof. Dr. Javier González-Gallego and Prof. Dr. Jesús R. Huerta

We thank all reviewers and editors for proving us fruitful comments and for the chance to revise our manuscript. We are also thankful to the reviewer’s efforts for spending their valuable time in reading the manuscript. These suggestive comments from reviewers are very convincing. Authors are providing answers to all the comments by point to point raised by reviewers’ comments. In addition, we received English editing by MDPI author services (https://www.mdpi.com/authors/english.).

Reviewer 2
In their manuscript by Monno et al. have investigated the ameliorating effect of exercise on diabetes kidney. The in vivo data clearly shows association between exercise, reduced body and fat weight, and renal function. In addition, change in oxidative parameters and glucose sensing proteins (mTOR and AMPK) were observed.
Overall, an interesting study and excellently presented, but some of the conclusions are not properly based on the results.

  • The authors' main objective was to investigate the molecular mechanism underlying the beneficial effect (abstract) and attributes it to the oxidative environment. However, only association evaluation is presented. An intervention with antioxidants would provide evidence. Here, the author also claims that AMPK activation and mTOR suppression is behind the reduced renal apoptosis (lines 234-240), which is suppressing as mTOR is typically an anti-apoptotic signal (by both main downstream complexes), leading me to a different conclusion - the alteration in AMPK, mTOR and oxidative markers are the results of the improvement and not mitigate its effect. Perhaps, all observations are due to increased insulin sensitivity?
    Ans)
    We thank reviewer’s evaluation and constructive comments.
    TET clearly resulted the improvement of kidney injuries including inflammation, oxidative stress and excessive apoptosis. In addition, impaired autophagy in WFRs was improved after TET, which was accompanied with AMPK activation and mTORC1 suppression. However, further studies are necessary to evaluate whether TET exerts the effects of anti-inflammation/oxidative stress and the reduced excess apoptosis through the restoration of autophagy via AMPK activation and mTORC1 suppression. Please also see answers against second your comment as below.
    The beneficial effects on kidney of TET were exerted in independent glucose levels, because there was no difference on the levels of HbA1c and blood glucose under ad libitum feeding between sedentary WFRs and WFRs with TET. Therefore, we wrote “In our study, TET reduced fat weights and increased gastrocnemius muscle weights, as observed in the WFRs-T, compared with the WFRs. However, HbA1c levels did not significantly differ between the WFRs with TET and sedentary WFRs. Thus, TET might provide a greater renoprotective effect independent of glucose levels.”, in the section of Discussion of first submitted manuscript. In addition, we added the sentences, “Previously, Gosh et al. also demonstrated that exercise using a motorized exercise wheel system for 1 hour every day in db/db mice improved diabetes-induced kidney injuries including renal cell excess apoptosis, without changes in plasma glucose or insulin status. In our study, results from intraperitoneal glucose and insulin tolerance test showed no difference between WFRs and WFRs with TET (data not shown). However, we did not evaluate insulin sensitivity by a glucose clamp technic, which has been recognized as a gold standard method for insulin sensitivity.”, in the section of Discussion. We also showed data regarding IPGTT and IPITT as below (Figure for reviewer). Finally, in the section of Conclusion, we added the sentence, “TET also might provide a greater renoprotective effect independent of glucose levels.”.

Figure for reviewer. IPGTT and IPITT: (a) Data from the intraperitoneal glucose tolerance test (IPGTT) and (c) intraperitoneal insulin tolerance test (IPITT), and (b) AUCs for the IPGTT and (d) IPITT at 44 weeks of age (IPGTT: n=4, IPITT: n=5). All data are means ± standard deviation (SD). ****p < 0.0001, ns: not significant. White circles, WLRs; black circles, WFRs; red triangles, WFRs-T. $$p < 0.01 vs WLRs and WFRs; $$$p < 0.001 vs WLRs and WFRs; \p < 0.05 vs WLRs and WFRs-T; \\p < 0.01 vs WLRs and WFRs-T; \\\p < 0.001 vs WLRs and WFRs-T. WLRs, sedentary Wistar lean rats; WFRs, sedentary Wistar fatty rats; WFRs-T, Wistar fatty rats with treadmill exercise training.

  • As DKD is highly complex, and the presented manuscript does advance the knowledge on the subject and the role of oxidative stress, the authors are recommended to revise their manuscript and change all causality to an association (between the reduced oxidative stress, AMPK and mTOR and the improved by TE) in the manuscript (including in the title and abstract). In addition, the discussion should also reflect the need for clarification.
    Ans)

We thank reviewer’s evaluation and constructive comments.
・We changed the title to “Exercise ameliorates diabetic kidney disease in type 2  

diabetic fatty rats”

・In Abstract, we changed the last sentence to “In conclusion, TET ameliorates diabetes-induced kidney injury in type 2 diabetic fatty rats.”.

・We changed the sentences in first paragraph in the section of Discussion to “In this study, TET ameliorated urinary albumin and L-FABP excretion as well as kidney injuries, including tubular cell damage and tubulointerstitial fibrosis, inflammation and oxidative stress, in WFRs. In addition, TET resulted in improvements by mitigating excessive apoptosis and restoring impaired autophagy in the renal cortex as well as mitochondrial morphology with respect to abnormalities in proximal tubular cells, in which were also accompanied by AMPK activation and mTORC1 suppression. The beneficial effects on kidney of TET were exerted independent of glucose levels, because there was no difference on the levels of HbA1c and blood glucose under ad libitum feeding between sedentary WFRs and WFRs with TET.

・We also added the sentences “However, further studies are necessary to clarify a causality among the TET-induced improvement of inflammation/oxidative stress, decrease in excess of apoptosis and the restoration of autophagy through AMPK activation and mTORC1 suppression, in the kidney of WFRs.” in third paragraph of Discussion.

3) Please also add in the abstract and dissection “male rats” or provide evidence that there is no difference in the results due to the gender

Ans)
We added “male rats” in the Abstract.

Reviewer 3 Report

This study examined the effects of TET on renal disease in a well-established model of T2DM (fa/fa). Results show that TET is beneficial and improved many of the markers typically associated with ESRD. Paper is well-written and results highlight the importance of modifiable factors in curbing the consequences of T2DM.

  1. In the Methods, can the authors justify the use of WLRs as control? This can be addressed.
  2. Include in this section (Methods), how a treadmill belt speed of 13m/min translates as VO2.
  3. Were the WLRs placed in the treadmill (with it turned off) along side the running rats during the exercise session? Again this needs to be added to the Methods.
  4. Were rats fasted during the tissue harvest? And how many hours after the last exercise session were the tissues collected?
  5. How was total fat weight determined? Was it the sum of the local fat deposits (suprarenal, visceral, etc/)? The local fat pad weights can be added to the results.
  6. Authors need to address why (in Discussion) fasting blood glucose levels were the same between control rats and fa/fa rats despite the clear differences in HB1ac levels.  
  7. Include the rationale for measuring FABP in urine of rats.
  8. Images in Figures 3-5 can be expanded for better view and the font size in all figures can be made clearer for readership.
  9. The benefits of ET on renal markers occurred in the absence of a reduction in blood glucose. This is an important observation that needs be discussed in the paper. Earlier work using the db/db mouse has reported these changes (after TET) occurring independently of blood glucose control. (See earlier work by I Laher, F Moien-Afshari, T Broderick in the db/db mouse). Describing potential mechanisms for the lack of reduction in glucose after TET should be in the Discussion.
  10. Hopefully, the authors will measure inflammatory cytokines to complement this work.

Author Response

Dear The Editors, Prof. Dr. Javier González-Gallego and Prof. Dr. Jesús R. Huerta

We thank all reviewers and editors for proving us fruitful comments and for the chance to revise our manuscript. We are also thankful to the reviewer’s efforts for spending their valuable time in reading the manuscript. These suggestive comments from reviewers are very convincing. Authors are providing answers to all the comments by point to point raised by reviewers’ comments. In addition, we received English editing by MDPI author services (https://www.mdpi.com/authors/english.).

Reviewer 3
This study examined the effects of TET on renal disease in a well-established model of T2DM (fa/fa). Results show that TET is beneficial and improved many of the markers typically associated with ESRD. Paper is well-written and results highlight the importance of modifiable factors in curbing the consequences of T2DM.

  1. In the Methods, can the authors justify the use of WLRs as control? This can be addressed.

Ans)

We thank reviewer’s comments. In this study, we aimed to evaluate the beneficial effect of exercise on diabetic and obese rats. WLRs were used as non-diabetic and non-exercise control. We inserted “(1) sedentary WLRs (WLRs, n = 8), non-diabetic and non-exercise control;” in the section of Material and Methods.

  1. Include in this section (Methods), how a treadmill belt speed of 13m/min translates as VO2.

Ans)

The intensity of the TET in this study is equivalent to approximately 50-60% of the maximal oxygen consumption (VO2max), as previously reported.

  1. Were the WLRs placed in the treadmill (with it turned off) along side the running rats during the exercise session? Again this needs to be added to the Methods.

Ans)

Sedentary WLRs and WFRs were placed on the nonmoving treadmill for 30 min for 5 days a week as non-exercise control.

  1. Were rats fasted during the tissue harvest? And how many hours after the last exercise session were the tissues collected?

Ans)

After 12 hours from the exercise session, we collected tissues sample under not fasting state.

  1. How was total fat weight determined? Was it the sum of the local fat deposits (suprarenal, visceral, etc/)? The local fat pad weights can be added to the results.

Ans)

Fat weight means abdominal fat weight including epididymal fat and retroperitoneal fat.

  1. Authors need to address why (in Discussion) fasting blood glucose levels were the same between control rats and fa/fa rats despite the clear differences in HB1ac levels.  

Ans)

The WFRs exhibit clearly high levels of blood glucose under ad libitum despite no change of fasting blood glucose levels between WFRs and WLRs, thereby, HbA1c levels were elevated in WFRs compared to those of WLRs.

  1. Include the rationale for measuring FABP in urine of rats.

Ans)

The L-FABP is recognized as the one of the biomarkers for renal tubular cell damage oxidative stress. Therefore, we added “liver-type fatty acid-binding protein (L-FABP; mouse/rat FABP1/L-FABP, R & D Systems, Inc., Minneapolis, MN, USA) which is the one of the biomarkers for renal tubular cell damage oxidative stress,” in the section of Materials and Methods.

  1. Images in Figures 3-5 can be expanded for better view and the font size in all figures can be made clearer for readership.

Ans)

We expanded all the Figures for better view and the font size.

  1. The benefits of ET on renal markers occurred in the absence of a reduction in blood glucose. This is an important observation that needs be discussed in the paper. Earlier work using the db/db mouse has reported these changes (after TET) occurring independently of blood glucose control. (See earlier work by I Laher, F Moien-Afshari, T Broderick in the db/db mouse). Describing potential mechanisms for the lack of reduction in glucose after TET should be in the Discussion.

Ans)

The beneficial effects on kidney of TET were exerted in independent glucose levels, because there was no difference on the levels of HbA1c and blood glucose under ad libitum feeding between sedentary WFRs and WFRs with TET. Therefore, we wrote “In our study, TET reduced fat weights and increased gastrocnemius muscle weights, as observed in the WFRs-T, compared with the WFRs. However, HbA1c levels did not significantly differ between the WFRs with TET and sedentary WFRs. Thus, TET might provide a greater renoprotective effect independent of glucose levels.”, in the section of Discussion of first submitted manuscript. In addition, we added the sentences, “Previously, Gosh et al. also demonstrated that exercise using a motorized exercise wheel system for 1 hour every day in db/db mice improved diabetes-induced kidney injuries including renal cell excess apoptosis, without changes in plasma glucose or insulin status. In our study, results from intraperitoneal glucose and insulin tolerance test showed no difference between WFRs and WFRs with TET (data not shown). However, we did not evaluate insulin sensitivity by a glucose clamp technic, which has been recognized as a gold standard method for insulin sensitivity.”, in the section of Discussion. We also showed data regarding IPGTT and IPITT as below (Figure for reviewer). Finally, in the section of Conclusion, we added the sentence, “TET also might provide a greater renoprotective effect independent of glucose levels.”.

Figure for reviewer. IPGTT and IPITT: (a) Data from the intraperitoneal glucose tolerance test (IPGTT) and (c) intraperitoneal insulin tolerance test (IPITT), and (b) AUCs for the IPGTT and (d) IPITT at 44 weeks of age (IPGTT: n=4, IPITT: n=5). All data are means ± standard deviation (SD). ****p < 0.0001, ns: not significant. White circles, WLRs; black circles, WFRs; red triangles, WFRs-T. $$p < 0.01 vs WLRs and WFRs; $$$p < 0.001 vs WLRs and WFRs; \p < 0.05 vs WLRs and WFRs-T; \\p < 0.01 vs WLRs and WFRs-T; \\\p < 0.001 vs WLRs and WFRs-T. WLRs, sedentary Wistar lean rats; WFRs, sedentary Wistar fatty rats; WFRs-T, Wistar fatty rats with treadmill exercise training.

  1. Hopefully, the authors will measure inflammatory cytokines to complement this work.

Ans)

We thank reviewer’s constructive comment and suggestion. We would like to measure inflammatory cytokines in not only the kidney but also in the blood and skeletal muscle and so on, in the next step of this study.

Round 2

Reviewer 1 Report

The authors have successfully answered all  comments and suggestions 

Reviewer 2 Report

all comments were addressed by the aoutors 

Reviewer 3 Report

The authors have addressed most of my comments. 

Removal of fat pads (abdominal, etc.) form rats can be added to the Methods (under experimental protocol).

Revise paper for minor spelling errors.

Author Response

We thank reviewer’s comments again.

We inserted the sentences regarding tissues collecting “At the same time, the perirenal, retroperitoneal and epididymal fat was dissected respectively, and fat weight was defined the sum of their fat pads, in this study. Gastrocnemius and soleus muscles also removed. After 12 hours from the exercise session, we collected tissues sample under not fasting state.”, in the section of Material and Methods.

In addition, we received English editing by MDPI author services (https://www.mdpi.com/authors/english.), and we edited our manuscript again.
